# Revisiting the Noise Model of Stochastic Gradient Descent

## Abstract

The stochastic gradient noise (SGN) is a significant factor in the success of stochastic gradient descent (SGD). Following the central limit theorem, SGN was initially modeled as Gaussian, and lately, it has been suggested that stochastic gradient noise is better characterized using $S\alpha S$ Lévy distribution. This claim was allegedly refuted and rebounded to the previously suggested Gaussian noise model. This paper presents solid, detailed empirical evidence that SGN is heavy-tailed and better depicted by the $S\alpha S$ distribution. Furthermore, we argue that different parameters in a deep neural network (DNN) hold distinct SGN characteristics throughout training. To more accurately approximate the dynamics of SGD near a local minimum, we construct a novel framework in $\mathbb{R}^N$, based on Lévy-driven stochastic differential equation (SDE), where one-dimensional Lévy processes model each parameter in the DNN. Next, we study the effect of learning rate decay (LRdecay) on the training process. We demonstrate theoretically and empirically that its main optimization advantage stems from the reduction of the SGN. Based on our analysis, we examine the mean escape time, trapping probability, and more properties of DNNs near local minima. Finally, we prove that the training process will likely exit from the basin in the direction of parameters with heavier tail SGN. We will share our code for reproducibility.

## 1 Introduction

The tremendous success of deep learning (Bengio, 2009; Hinton et al., 2012; LeCun et al., 2015) can be partly attributed to implicit properties of the optimization tools, in particular, the popular SGD (Robbins & Monro, 1951; Bottou, 1991) scheme. Despite its simplicity, i.e., being a noisy first-order optimization method, SGD empirically outperforms gradient descent (GD) and second-order methods. The stochastic gradient noise of stochastic gradient descent can improve the model's generalization by escaping from sharp basins and settling in wide minima (Ziyin et al., 2021; Smith et al., 2020). SGD noise stems from the stochasticity in the mini-batch sampling operation, whose formation and amplitude are affected by the DNN architecture and data distribution. The main hurdle in improving deep learning is the lack of theory behind specific processes and modules frequently used; better understanding will help break current barriers in the field. Hence, studying the properties of SGD should be of the highest priority. Analyzing the behavior of SGD optimization for non-convex cost functions, is an ongoing research (Chaudhari & Soatto, 2018; Zhou et al., 2019; Draxler et al., 2018; Nguyen & Hein, 2017; He et al., 2019b; Li et al., 2017; Smith et al., 2021; Ziyin et al., 2021; You et al., 2019). The problem of analyzing SGD noise has recently received much attention. Studies mainly examine the distribution and nature of the noise, with its ability to escape local minima and generalize better (Hu et al., 2017; He et al., 2019a; Wu et al., 2019; HaoChen et al., 2020; Zhou et al., 2019; Keskar et al., 2016).

SGD is based on an iterative update rule; the $k - th$ step of that iterative update rule is formulated as follows:

$$w_k = w_{k-1} - \frac{\eta}{B} \sum_{\ell \in \Omega_t} \nabla U^{(\ell)}(w_{k-1}) = w_{k-1} - \eta \nabla U(w_{k-1}) + \epsilon u_k, \tag{1}$$

where $w$ denotes the weight (parameters) of the DNN, $\nabla U(w)$ is the gradient of the objective function, $B$ is the batch size, $\Omega_k \subset \{1, .., D\}$ is the randomly selected mini-batch, thus $|\Omega_k| = B$, $D$ is the number of data

points in the dataset, $u_k$ is the SGD noise and it is formulated as: $u_k = \nabla U(w_k) - \frac{1}{B}\sum_{\ell \in \Omega_k} \nabla U^{(\ell)}(w_k)$, i.e. the difference between the gradient produced by GD and SGD, finally $\epsilon = \eta^{\frac{\alpha-1}{\alpha}}$, and $\eta$ is the learning rate.

As gradient flow is a popular apparatus to understand GD dynamics, continuous-time SDE is used to investigate the SGD optimization process and examine the time evolution of the dynamic system in the continuous domain (Zhu et al., 2018; Meng et al., 2020; Xie et al., 2020; Chaudhari & Soatto, 2018; Hu et al., 2017; Sato & Nakagawa, 2014a).

Empiric experiments and their results produced a lively discussion on how SGN distributes, most of previous works (Zhu et al., 2018; Mandt et al., 2016; Wu et al., 2020; Ziyin et al., 2021): argue that the noise is Gaussian , i.e. $u_t \sim \mathcal{N}(0, \lambda(w_t))$, where $\lambda(w_t)$ is the noise covariance matrix and formulated as follows:

$$\lambda(W_t) = \frac{1}{B}\left[\frac{1}{D}\sum_{j=1}^{D}\nabla U^{(j)}(W_t)\nabla U^{(j)}(W_t)^T - \nabla U(W_t)\nabla U(W_t)^T\right]. \tag{2}$$

Recently, Zhu et al. (2018) showed the importance of modeling the SGN as an anisotropic noise to more accurately approximate SGD's dynamics. In Simsekli et al. (2019) the authors argue that SGN obeys $\mathcal{S}\alpha\mathcal{S}$ Lévy distribution, due to SGN's heavy-tailed nature. $\mathcal{S}\alpha\mathcal{S}$ Lévy process is described by a single parameter $\alpha_i$, also named "stability parameter," and holds unique properties, such as large discontinuous jumps. Therefore, Lévy-driven SDE does not depend on the height of the potential; on the contrary, it directly depends on the horizontal distance to the domain's boundary; this implies that the process can escape from narrow minima – no matter how deep they are and will stay longer in wide minima. In this work, we claim that the noise of different parameters in the DNN distributes differently and argue that it is crucial to incorporate this discrepancy into the SGN model. Hence, we model the training process as Lévy-driven stochastic differential equations (SDEs) in $\mathbb{R}^N$, where each parameter $i$ distributes with a unique $\alpha_i$; this formulation helps us investigate the properties and influence of each parameter on the training process.

Another critical aspect of NN optimization is the learning rate. Bengio (2012) argue that the learning rate is "the single most important hyper-parameter" in training DNNs; we yearn to understand what role the LRdeacy has in SGN. Therefore, we examine the effect of the learning rate scheduler on the training process; considering two schedulers, the exponential scheduler $s_t = t^{\gamma-1}$ and the multi-step scheduler using $p$ training phases with $p$ different factors: $s_t = \gamma_p, \forall t \in (T_p, T_{p+1}]$, s.t $\gamma_p \in (0,1)$ , the first is analysed for better theoretical reasoning, the last is a popular discrete scheduler used in modern DNNs training. We argue that decreasing the learning rate helps the optimization by attenuating the noise and not by reducing the step size; we brace the above claim using theoretical and experimental evidence.

Our contributions can be summarized as follows:

- This work empirically shows that the SGN of each parameter in a deep neural network is better characterized by $S\alpha S$ distribution.

- Our experiments strongly indicate that different parametric distributions characterize the noise of distinct parameters.

- We propose a novel dynamical system in $\mathbb{R}^N$ consisting of $N$ one-dimensional Lévy processes with $\alpha_i$-stable components and incorporates a learning rate scheduler to depict the training process better.

- Using our framework, we present an approximation of the mean escape time, the probability of escaping the local minima using a specific parameter, and more properties of the training process near the local minima.

- We prove that parameters with lower $\alpha_i$ hold more probability to aid the training process to exit from the local minima.

- We show that the effectiveness of the learning rate scheduler mainly evolves from noise attenuation and not step decaying.

## 2 Related Work

The study of stochastic dynamics of systems with small random perturbations is a well-established field, first by modeling as Gaussian perturbations (Freidlin et al., 2012; Kramers, 1940), then replaced by Lévy noise with discontinuous trajectories (Imkeller & Pavlyukevich, 2006a; Imkeller et al., 2010; Imkeller & Pavlyukevich, 2008; Burghoff & Pavlyukevich, 2015). Modeling the noise as Lévy perturbations has attracted interest in the context of extreme events modeling, such as in climate (Ditlevsen, 1999), physics (Brockmann & Sokolov, 2002) and finance (Scalas et al., 2000).

**Remark** Let us remind that a Symmetric $\alpha$ stable distribution ($S\alpha S$ or Lévy $S\alpha S$) is a heavy-tailed distribution, parameterized by $\alpha$ - the stability parameter, smaller $\alpha$ leads to heavier tail (i.e., extreme events are more frequent and with more amplitude), and vice versa.

Modeling SGD using differential equations is a deep-rooted method, (Li et al., 2015) showed a framework of SDE approximation of SGD and focused on momentum and adaptive parameter tuning schemes and the dynamical properties of those stochastic algorithms. (Mandt & Blei, 2015) employed a similar procedure to derive an SDE approximation for the SGD to study issues such as choice of learning rates. (Li et al., 2015) showed that SGD can be approximated by an SDE in a first-order weak approximation. The early works in the field of studying SGD noise have approximated SGD by Langevin dynamic with isotropic diffusion coefficients (Sato & Nakagawa, 2014b; Raginsky et al., 2017; Zhang et al., 2017), later more accurate modeling suggested (Mandt et al., 2017; Zhu et al., 2018; Mori et al., 2021) using an anisotropic noise covariance matrix. Lately, it has been argued (Simsekli et al., 2019) that SGN is better characterized by $S\alpha S$ noise, presenting experimental and theoretical justifications. This model was allegedly refuted by (Xie et al., 2020), claiming that the experiments performed by (Simsekli et al., 2019) are inaccurate since the noise calculation was done across parameters and not across mini-batches. Lévy driven SDEs Euler approximation literature is sparser than for the Brownian motion SDEs; however, it is still intensely investigated; for more details about the convergence of Euler approximation for Lévy discretization, see (Mikulevicius & Zhang, 2010; Protter et al., 1997; Burghoff & Pavlyukevich, 2015).

Learning rate decay is an essential technique in training DNNs, investigated first for gradient descent (GD) by (LeCun et al., 1998). Kleinberg et al. (2018) showed that SGD is equivalent to the convolution of the loss surface, with the learning rate serving as the conceptual kernel size of the convolution. Hence spurious local minima can be smoothed out; thus, the decay of the learning rate later helps the network converge around the local minimum. You et al. (2019) suggested that learning rate decay improves the ability to learn complex separation patterns.

## 3 Framework and Main Results

In our analysis, we consider a DNN with $\bar{\mathbf{L}}$ layers and a total of $N$ weights, the domain $\mathcal{G}$ is the local environment of a minimum, in this environment, the potential $U(W_t)$ can be approximated by second-order Taylor expansion and is $\mu-$strongly convex near the minimum $W^*$ in $\mathcal{G}$ (see Appendix B.2 to better understand this assumption). Our framework considers an $N$-dimensional dynamic system, representing the update rule of SGD as a Lévy-driven stochastic differential equation. In contrast to previous works (Zhou et al., 2020; Simsekli et al., 2019), our framework does not assume that SGN distributes the same for every parameter in the DNN. Thus, the SGN of each parameter is characterized by a different $\alpha$. The governing SDE that depicts the SGDs dynamic inside the domain $\mathcal{G}$ is as follows:

$$W_t = w - \int_0^t \nabla U(W_p)\, dp + \sum_{l=1}^N s_t^{\frac{\alpha_l - 1}{\alpha_l}} \epsilon \mathbf{1}^T \lambda_l(t) r_l L_t^l, \tag{3}$$

where $W_t$ is the process that depicts the evolution of DNN weights while training, $L_t^l \in \mathbb{R}$ is a mean-zero $S\alpha S$ Lévy processes with a stable parameter $\alpha_l$. $\lambda_l(t) \in \mathbb{R}^N$ is the $l$-th row of the noise covariance matrix, $\mathbf{1} \in \mathbb{R}^N$ is a vector of ones, and its purpose is to sum the $l$-th row of the noise covariance matrix. $r_l \in \mathbb{R}^N$ is a unit vector and we demand $|\langle r_i, r_j \rangle| \neq 1$, for $i \neq j$, we will use $r_i$ as a one-hot vector although it is not necessary. $s_t$ will describe the learning rate scheduler, and $w$ are the initial weights.

**Remark** $L^l$ can be decompose into a small jump part $\xi_t^l$, and an independent part with large jumps $\psi_t^l$, i.e $L_l = \xi_t^l + \psi_t^l$, more information on $S\alpha S$ process appears in A.2.

Let $\sigma_{\mathcal{G}} = \inf\{t \geq 0 : W_t \notin \mathcal{G}\}$ depict the first exit time from $\mathcal{G}$. $\tau_k^l$ denotes the time of the $k$-th large jump of parameter $l$ driven by $\psi^l$ process, where we define $\tau_0 = 0$. The interval between large jumps is denoted as: $S_k^l = \tau_k^l - \tau_{k-1}^l$ and is exponentially distributed with mean $\beta_i(t)^{-1}$, while $\tau_k^l$ is gamma distributed $Gamma(k, \beta_l(t))$; $\beta_l(t)$ is the jump intensity and will be defined in Sec 3.2. We will define the arrival time of the $k$-th jump of all parameters combined as $\tau_k^*$, for $k \geq 1$ by

$$\tau_k^* \triangleq \bigwedge_{\tau_j^l > \tau_{k-1}^*} \tau_j^l, \tag{4}$$

following that $S_k^* = \tau_k^* - \tau_{k-1}^*$.

**Notations** In what follows, an upper subscript denotes the DNN's parameter index, while a lower subscript denotes time if $t$ is written or the discrete jump index unless it is specifically mentioned otherwise.

Jump heights are notated as: $J_k^l = \psi_{\tau_k}^l - \psi_{\tau_{k-}}^l$. We will define $\alpha_\nu$ as the average $\alpha$ parameter over the entire DNN; this will help us describe the global properties of our network.
Let us define a measure of horizontal distance from the domain boundary using $d_i^+$ and $d_i^-$; additional assumptions and a rigorous formulation of the assumptions can be found in Sec. G.
We will define two more processes to understand better the dynamics inside the basin (between the large jumps).

**The deterministic process** denoted as $Y_t$ is affected by the drift alone, without any perturbations. This process starts within the domain and does not escape this domain as time proceeds. The drift forces this process towards the stable point $W^*$ as $t \to \infty$, i.e., the local minimum of the basin; furthermore, the process converges to the stable point exponentially fast and is defined for $t > 0$, and $w \in \mathcal{G}$ by:

$$Y_t = w - \int_0^t \nabla U(Y_s)\, ds. \tag{5}$$

The following Lemma shows how fast $Y_t$ converges to the local minima from any starting point $w$ inside the domain.

**Lemma 3.1.** $\forall w \in \mathcal{G}$ , $\tilde{U} = U(w) - U(W^*)$, the process $Y_t$ converges to the minimum $W^*$ exponentially fast:

$$\|Y_t - W^*\|^2 \leq \frac{2\tilde{U}}{\mu} e^{-2\mu t}. \tag{6}$$

*See the proof Appendix D.6*

**The small jumps process** $Z_t$ composed from the deterministic process $Y_t$ and a stochastic process with infinite small jumps denoted as $\xi_t$ (see more details in A.2). $Z_t$ describes the system's dynamic in the intervals between the large jumps; hence we add an index $k$ that represents the index of the jump. Due to strong Markov property, $\xi_{t+\tau}^l - \xi_\tau^l, t \geq 0$ is also a Lévy process with the same law as $\xi^l$. Hence, for $t \geq 0$ and $k \geq 0$:

$$\xi_{t,k}^l = \xi_{t+\tau_{k-1}}^l - \xi_{\tau_{k-1}}^l. \tag{7}$$

The full small jumps process for $\forall t \in [0, S_k]$ is defined as:

$$Z_{t,k} = w + \int_0^t \nabla U(y_s)ds + \sum_{l=1}^N s_t^{\frac{\alpha_l - 1}{\alpha_l}} \epsilon \mathbf{1}^T \lambda_l(t) r_l \xi_{t,k}^l. \tag{8}$$

In the following proposition, we estimate the deviation of the solutions of the SDE driven by the process of the small jumps $Z_{t,k}^l$ from the deterministic trajectory in the $l$-th axis:

**Proposition 3.2.** *Let $T_\epsilon > 0$ exponentially distributed with parameter $\beta_l$ , $\forall w \in \mathcal{G}$, $c > 0$ and $\bar{\theta}_l \triangleq -\rho(1 - \alpha_l) + 2 - 2\theta_l$ , s.t $\theta_l \in (0, \frac{2 - \alpha_l}{4})$, and $C_{\theta_l} > 0$, s.t. the following holds:*

$$P\left( \sup_{t \in [0, T_\epsilon]} |Z_{t,k}^l(w) - Y_{t,k}^l(w)| \geq c\bar{\epsilon}^{-\theta_l} \right) \leq C_{\theta_l} \bar{\epsilon}^{\bar{\theta}_l} \tag{9}$$

Let us remind that: $\bar{\epsilon}_l = s_t^{\frac{\alpha_l - 1}{\alpha_l}} \epsilon_l$. In plain words, proposition 3.2 describes the distance between the deterministic process and the process of small jumps, between the occurrences of the large jumps. It indicates that between the large jumps, the processes are close to each other with high probability. The proof appears in D.3.

Next, we turn our attention to another property of the process of the small jumps $Z_{t,k}^l$. This will help us in understand the noise covariance matrix. First, we present additional notations: $H()$ and $\nabla U$ are the hessian and the gradient of the objective function. To denote different mini-batches, we use subscript $d$. That is, $H_d()$ and $\nabla U_d(W^*)$ are the hessian and gradient of the $d$ mini-batch. To represent different parameter, $u_{d,l}$ is the gradient of the $l$-th parameter, after forward pass of mini-batch $d$. Furthermore, $h_{l,j}$ represents the $i$-th row and $j$-th column of $H(W^*)$ which is the hessian after a forward pass of the entire dataset $D$, i.e., the hessian when performing standard Gradient Descent. Using stochastic asymptotic expansion, we are able to approximate $Z_{t,k}^l$ using the deterministic process and a first-order approximation of $Z_{t,k}^l$.

**Lemma 3.3.** *For a general scheduler $s_t$, let $\mu_\xi^i = 2t\left[ \frac{\bar{\epsilon}^{-\rho(1 - \alpha_l)} - 1}{1 - \alpha_l} \right]$, $\rho \in (0, 1)$ ,$\bar{\epsilon}_l = s_t^{\frac{\alpha_l - 1}{\alpha_l}} \epsilon_l$, $h_{ll}$ the second derivative of the drift in the $l$-th direction, $\forall w_l, w_j \in \mathcal{G}$, starting point after a big jump at time $\tau_k^* + p$ where $p \to 0$, and $A_{lj}(t) \triangleq \bar{\epsilon}_l w_j e^{-h_{jj}t} \mu_\xi^l (2t + \frac{1}{h_{ll}}(1 - e^{-h_{ll}t}))$, for $t \in [0, S_k^*)$ the following fulfills:*

$$\mathbb{E}[Z_{t,k}^l Z_{t,k}^j] \approx w_l w_j e^{-(h_{ll} + h_{jj})t} + A_{jl}(t) + A_{lj}(t) \tag{10}$$

Lemma 3.3 depicts the dynamics between two parameters in the intervals between the large jumps; this will aid in expressing the covariance matrix more explicitly; please examine the complete derivation of this result in the Appendix D.4.

### 3.1 Noise covariance matrix

The covariance of the noise matrix holds a vital role in modeling the training process; in this subsection, we aim to achieve an expression of the noise covariance matrix based on the stochastic processes we presented in the previous subsection. We can achieve the following approximation using stochastic Taylor expansion near the basin $W^*$.

**Proposition 3.4.** *Let us define $\tilde{u}_l = \sum_{j=1}^N \nabla u_l \nabla u_j$, $\tilde{h}_{l,m,p,j} := \frac{1}{B} \sum_{b=1}^B h_{b,l,m} h_{b,p,j}$, $h_{l,m,p,j} := h_{l,m} h_{p,j}$ and $\bar{h}_{l,m,p,j} := \tilde{h}_{l,m,p,j} - h_{l,m,p,j}$, then for any $t \in [0, S_k^*)$, the sum of the $l$-th row of the covariance matrix:*

$$\mathbf{1}^T \lambda_l^k(W_t) \approx \frac{1}{D} \sum_{j=1}^N \bar{u}_{lj} + \sum_{j=1}^N \sum_{m=1}^N \sum_{p=1}^N \bar{h}_{l,m,p,j}(w_m w_p e^{-(h_{mm} + h_{pp})t} + A_{mp}(t) + A_{pm}(t)), \tag{11}$$

where $A_{mp}(t)$ and $A_{pm}(t)$ are defined in lemma 3.3, see proof in Appendix D.5.

### 3.2 Jump Intensity

Let us denote $\beta_l(t)$ as the jump intensity of the compound Poisson process $\xi_l$, $\beta_l(t)$ define how high the process will jump and the frequency of the jumps, which are distributed according to the law $\beta_l(t)^{-1} \nu_\eta$, and the jump intensity is formulated as:

$$\beta_l(t) = \nu_{\eta_l}(\mathbb{R}) = \int_{\mathbb{R}/[-O,O]} \nu_l(dy) = \frac{2}{\alpha_i} s_t^{\rho(\alpha_l - 1)} \epsilon_l^{\rho \alpha_l}, \tag{12}$$

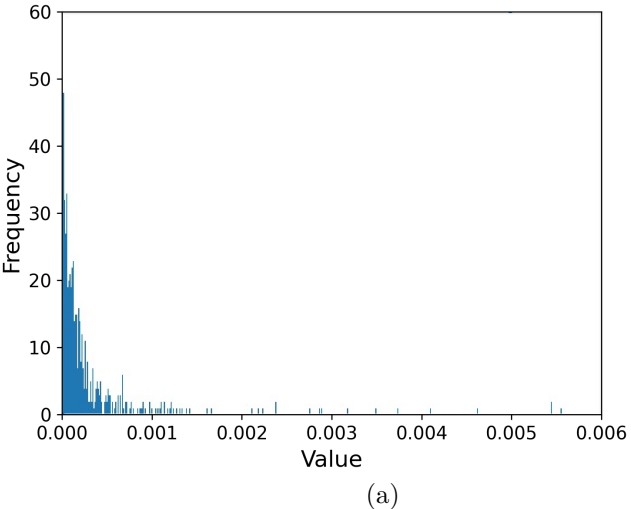 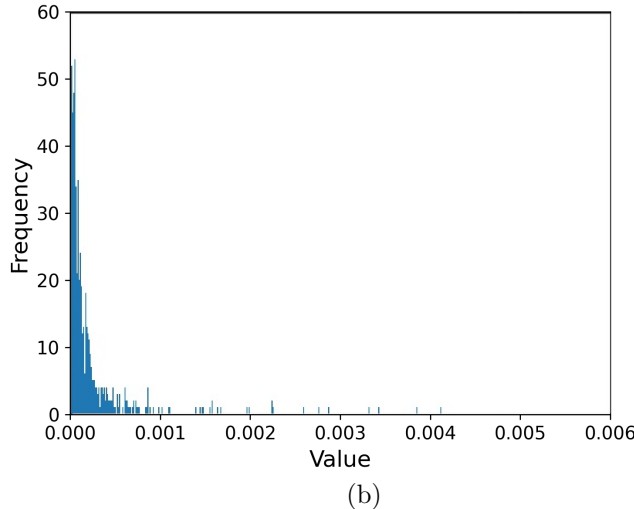

(a)  (b)

Figure 1: Histograms of the stochastic gradient noise for a single parameter. The top row shows the noise frequencies in ResNet34 for :(a) layer number 1, (b) layer number 2

| Model | Gauss-SSE | $S\alpha S$-SSE | Gauss-Chi2 | $S\alpha S$-Chi2 |
|---|---|---|---|---|
| ResNet110 | $7.43 \pm 9.14$ | $7.31 \pm 9.15$ | $2.38 \pm 1.61$ | $2.32 \pm 1.49$ |
| ResNet18 | $120.44 \pm 145.52$ | $87.43 \pm 130.91$ | $7.78 \pm 5.15$ | $5.87 \pm 4.75$ |
| ResNet34 | $36.47 \pm 41.27$ | $29.66 \pm 35.22$ | $4.46 \pm 3.21$ | $3.64 \pm 2.77$ |
| ResNet50 | $288.58 \pm 417.24$ | $203.70 \pm 331.19$ | $10.78 \pm 9.13$ | $8.01 \pm 7.34$ |

Table 1: The fitting error between SGN and $S\alpha S$/Gaussian distribution. Averaged over 100 randomly sampled parameters, four different CNNs trained on CIFAR100 with a batch size of 400. Two measures were used to evaluate the fitting error of each distribution, Sum of Squares Error (SSE) and Chi-Squares (Chi2). "Gauss" represents the Gaussian distribution. Our results demonstrate that $S\alpha S$ better depicts SGN.

where the integration boundary is $O \triangleq \epsilon^{-\rho} s_t^{-\rho \frac{\alpha_l - 1}{\alpha_l}}$, which is time-dependent, since the jump intensity is not stationary. The jump intensity is not stationary due to the learning rate scheduler, which decreases the size and frequency of the large jumps.

Let us denote $\beta_S(t) \triangleq \sum_{l=1}^{N} \beta_l(t)$, to ease the calculation we will assume that the time dependency: $\frac{\beta_l(t)}{\beta_S(t)} = \frac{\bar{\beta}_l}{\bar{\beta}_S} s^{\rho(\alpha_l - \alpha_\nu)}$. The probability of escaping the local minima in the first jump, in a single axis perspective, is denoted as:

$$P(s_t \epsilon \mathbf{1}^T \lambda_l(t) J_1^l \notin [d_l^-, d_l^+]) = \frac{m_l(t) \Phi_l s_t^{\alpha_l - 1}}{\beta_l(t)}, \tag{13}$$

where $m_l(t) = \frac{\mathbf{1}^T \lambda_l(t) \epsilon_l^{\alpha_l}}{\alpha_l}$, and $\Phi_l = (-d_l^-)^{-\alpha_l} + (d_l^+)^{-\alpha_l}$.

## 4 Theorems

In the following section, we provide a theoretical analysis of SGD dynamics during the training of DNNs. Our analysis is based on two empirical pieces of evidence demonstrated in this work; the first is that SGN is indeed heavy-tailed. The second is that each parameter in the DNN's training process has a different stability parameter $\alpha$ drastically affects the noise properties.

Our work will assume that the training process can exit from the domain only at times that coincide with large jumps. This assumption is based on a few realizations; first, the deterministic process $Y_t$ initialized in any point $w \in \mathcal{G}_\delta$, will converge to the local minima of the domain (see G 3). Second, $Y_t$ converges to the minimum much faster than the average temporal gap between the large jumps; third, using lemma 3.1 we understand that the small jumps are less likely to help the process escape from the local minimum. Next, we will show evidence for the second realization mentioned above, the relaxation time $T_R^l$ is the time for the deterministic process $Y_t^l$, starting from any arbitrary $w \in \mathcal{G}$, to reach an $\bar{\epsilon}_l^\zeta$-neighbourhood of the attractor. For some $C_1 > 0$, the relaxation time is

$$T_R^l = \max \left\{ \int_{d_l^-}^{-\bar{\epsilon}_l^\zeta} \frac{dy}{-U'(y)_l}, \int_{\bar{\epsilon}_l^\zeta}^{d_l^+} \frac{dy}{U'(y)_l} \right\} \leq C_1 |ln\bar{\epsilon}_l|. \tag{14}$$

Now, let us calculate the expectation of $S_k^* = \tau_k^* - \tau_{k-1}^*$, i.e. the interval between the large jumps:

$$\mathbb{E}[S_k^l] = \mathbb{E}[\tau_k^l - \tau_{k-1}^l] = \beta_l^{-1} = \frac{\alpha_l}{2} s_t^{-\rho\alpha_l} \epsilon^{-\rho\alpha_l}. \tag{15}$$

It is easy to notice that $\mathbb{E}[S_k^l] \gg T_R$, thus we can approximate that the process $W_t$ is near the neighborhood of the basin, right before the large jumps. This means that it is highly improbable that two large jumps will occur before the training process returns to a neighborhood of the local minima. Using the information described above, we analyze the escaping time for the exponential scheduler and for the multi-step scheduler; expanding our framework for more LRdecay schemes is straightforward. Let us define a constant that will be used for the remaining of the paper: $A_{l,\nu} \triangleq (1 - \bar{m}_\nu \bar{\beta}_\nu^{-1} \Phi_\nu)(1 - \bar{\beta}_l \bar{\beta}_S^{-1})$, for the next theorem we denote: $C_{l,\nu,p} \triangleq \frac{2+(\gamma-1)(\alpha_l-1+\rho(\alpha_l-\alpha_\nu))}{1+(\gamma-1)(\alpha_l-1)}$, where $C_{l,\nu,p}$ depends on $\alpha_l$, $\gamma$, and on the difference $\alpha_l - \alpha_\nu$. The following theorem describes the approximated mean transition time for the exponential scheduler:

**Theorem 4.1.** *Given $C_{l,\nu,p}$ and $A_{l,\nu}$, let $s_t$ be an exponential scheduler $s_t = t^{\gamma-1}$, the mean transition time from the domain $\mathcal{G}$:*

$$\mathbb{E}[\sigma_\mathcal{G}] \leq \sum_{l=0}^N A_{l,\nu}^{-1} \frac{\beta_l(\bar{m}_l \Phi_l)^{1-C_{l,\nu,p}}}{\beta_S(1+(\gamma-1)(\alpha_l-1))} \Gamma(C_{l,\nu,p})$$

Where $\Gamma$ is the gamma function, $\bar{m}_l = \frac{\bar{\lambda}_l^{\alpha_l} \epsilon_l^{\alpha_l}}{\alpha_l}$ and $\bar{\beta}_l = \frac{2\epsilon_l^{\rho\alpha_l}}{\alpha_l}$ is the time independent jump intensity. For the full proof, see D.1. It can be easily observed from Thm. 4.1 that as $\gamma$ decreases, i.e., faster learning rate decay, the mean transition time increases. Interestingly, when $\alpha_l \to 2$ (nearly Gaussian) and $\gamma \to 0$, the mean expectation time goes to infinity, which means that the training process is trapped inside the basin.

**Corollary 4.2.** *Using Thm. 4.1, if the cooling rate is negligible, i.e $\gamma \to 1$, the mean transition time:*

$$\mathbb{E}[\sigma_\mathcal{G}] \leq \sum_{l=0}^N A_{l,\nu}^{-1} \frac{1}{\beta_S 1^T \bar{\lambda}_l \epsilon^{\alpha_l(1-\rho)} \Phi_l}. \tag{16}$$

By expanding $\bar{\lambda}_l$ and $\beta_S$ in 16, we can express the mean escape time as a function of the learning rate $\eta$; hence we can obtain the following term:

$$\mathbb{E}[\sigma_\mathcal{G}] \leq \sum_{l=0}^N \Phi_l^{-1} A_{l,\nu}^{-1} \frac{1}{\sum_{i=1}^N C_{l,i} \eta^{\frac{\alpha_l-\alpha_i}{2}} + C_{j\lambda} \sum_{i,k,p=1}^N (\eta^{1+\frac{\alpha_l-\alpha_i}{2}+\frac{\alpha_p-1}{\alpha_p}} w_p + \eta^{1+\frac{\alpha_l-\alpha_i}{2}+\frac{\alpha_k-1}{\alpha_k}} w_k)} \tag{17}$$

One can observe that the denominator is a linear combination of polynomials that depends on the learning rate and the $\alpha$ value of each parameter. $C_{l,i}$ is a function of the Hessian gradient, and the weights of the DNN, $C_{j\mu}$ depends on the Hessian, weights, and the mean of the small jump process; for further details D.1.

The framework presented in this work enables us to understand in which direction $r_i$ the training process is more probable to exit the basin $\mathcal{G}$, i.e., which parameter is more liable to help the process escape; this is a crucial feature for understanding the training process. The following theorems will be presented for the exponential scheduler but can be expanded for any scheduler.

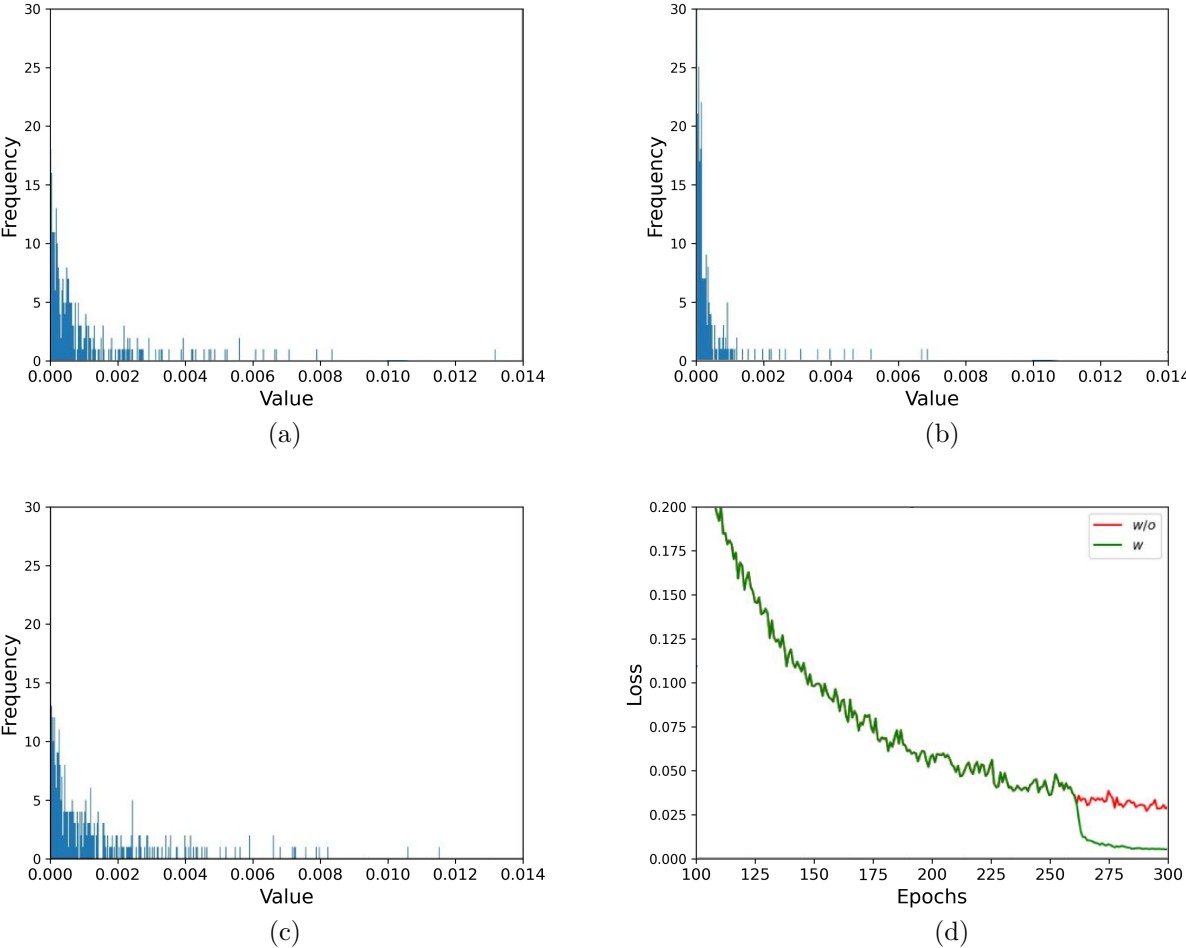

Figure 2: The stochastic gradient noise of a single parameter in ResNet110 He et al. (2015). (a) Before applying learning rate decay, at epoch 279. (b) After applying learning rate decay, at epoch 281. (c) Without learning rate decay, at epoch 281. (d) The training loss with and without learning rate decay applied at epoch 280.

**Theorem 4.3.** *Let $s_t$ be an exponential scheduler $s_t = t^{\gamma-1}$, $C_l \triangleq \frac{(\gamma-1)(\alpha_l - 1 + \rho(2\alpha_l - \alpha_\nu - \alpha_l)) + 2}{(\gamma-1)(\alpha_l - 1) + 1}$, for $\delta \in (0, \delta_0)$, the probability of the training process to exit the basin through the l-th parameter is as follows:*

$$P(W_\sigma \in \Omega_i^+(\delta)) \leq \sum_{l=0}^{N} A_{l,\nu}^{-1} \frac{\bar{m}_i \Phi_i}{\bar{\beta}_i} (d_i^+)^{-\alpha_i} \frac{\beta_l^2 (\bar{m}_l \Phi_l)^{-C_l}}{\beta_S((\gamma-1)(\alpha_l - 1) + 1)} \Gamma(C_l). \tag{18}$$

Let us focus on the terms that describes the $l$-th parameter:

$$P(W_\sigma \in \Omega_l^+(\delta)) \leq \frac{\bar{m}_i}{\bar{\beta}_i} (d_i^+)^{-\alpha_i} \sum_{l=0}^{N} \tilde{C}_l, \tag{19}$$

where $\tilde{C}_l$ encapsulate all the terms that do not depend on $i$. When considering SGN as Lévy noise, we can see that the training process needs only polynomial time to escape a basin. The following result helps us to assess the escaping ratio of two parameters.

| Model | BS | Gauss-SSE | $S\alpha S$-SSE | Gauss-Chi2 | $S\alpha S$-Chi2 |
|---|---|---|---|---|---|
| Bert Base | 8 | $2.15 \pm 0.64$ | $0.71 \pm 0.33$ | $1.41 \pm 0.43$ | $0.42 \pm 0.21$ |
| Bert Base | 32 | $0.37 \pm 0.33$ | $0.18 \pm 0.12$ | $0.09 \pm 0.07$ | $0.11 \pm 0.06$ |

Table 2: The fitting errors. The errors were computed by averaging 120 randomly sampled parameters from BERT Devlin et al. (2018) model trained on the Cola dataset. We use two measures to evaluate the fitting error of each distribution, Sum of Squares Error (SSE) and Chi-Squares (Chi2). Gauss represents the Gaussian distribution.

**Corollary 4.4.** *The ratio of probabilities for exiting the local minima from two different DNN parameters is:*

$$\frac{P(W_\sigma \in \Omega_l^+(\delta))}{P(W_\sigma \in \Omega_j^+(\delta))} \le \frac{1^T \lambda_l^{\alpha_l}}{1^T \lambda_j^{\alpha_j}} \epsilon^{(\alpha_l - \alpha_j)(1-\rho)} \frac{(d_l^+)^{-\alpha_l}}{(d_j^+)^{-\alpha_j}} \tag{20}$$

Let us remind the reader that $(d_i^+)$ is a function of the horizontal distance from the domain's edge. Therefore, the first conclusion is that the higher $(d_l^+)$ is, the probability of exiting from the $l$-th direction decreases. However, the dominant term is $\epsilon^{(\alpha_l - \alpha_j)(1-\rho)}$, combining both factors, parameters with lower $\alpha$ will have more chance of being in the escape path. It can also be seen from the definition of $\beta_l$ that parameters with lower $\alpha$ jump earlier and contribute more significant jump intensities. We can conclude by writing:

$$\frac{P(W_\sigma \in \Omega_l^+(\delta))}{P(W_\sigma \in \Omega_j^+(\delta))} \propto \epsilon^{\Delta_{l,j}}, \tag{21}$$

where $\Delta_{l,j} = \alpha_l - \alpha_j$.

The next Theorem will evaluate the probability of exiting the basin after time $u$.

**Theorem 4.5.** *Let $s_t = t^{\gamma-1}$, where $\gamma$ is the cooling rate; let us denote two constants that express the effect of the scheduler: $\gamma_l \triangleq 1 + (\gamma-1)(\alpha_l - 1)$ and $\kappa \triangleq \frac{1+(\gamma-1)(\alpha_l - 1 + \rho(\alpha_l - \alpha_\nu))}{\gamma_l}$, for $u > 0$:*

$$P(\sigma > u) \le \sum_{l=0}^{N} A_{l,\nu}^{-1} \frac{\bar{\beta}_l \bar{m}_l \Phi_l}{\bar{\beta}_S \gamma_l (\bar{m}_l \Phi_l)^\kappa} \Gamma \left( \kappa, \bar{m}_l \Phi_l u^{\gamma_l} \right) \quad . \tag{22}$$

**Corollary 4.6.** *Using Thm. E.4, for $\gamma \to 1$:*

$$P(\sigma > u) \le \sum_{l=0}^{N} A_{l,\nu}^{-1} \frac{\bar{\beta}_l}{\bar{\beta}_S} e^{-\bar{m}_l \Phi_l u} \quad . \tag{23}$$

4.6 shows that the probability of exiting a basin after $u$ iterations decays exponentially with respect to $u$, $\bar{m}_l$, and $\Phi_l$. The value $\Phi_l$ describes the horizontal width of the basin, and $\bar{m}_l$ is a function of the learning rate and the noise covariance matrix.

## 5 Experiments

This section presents the core experimental results supporting our analysis; additional experiments can be found in the Appendix. All the experiments were conducted using SGD without momentum and weight decay.

**Stochastic gradient noise distribution** We empirically show that SGN is better characterized using the $S\alpha S$ Lévy distribution. Unlike previous works (Simsekli et al., 2019; Zhou et al., 2020; Xie et al., 2020) we use numeric results to demonstrate the heavy-tailed nature of SGN. Our methodology follows Xie et al. (2020),

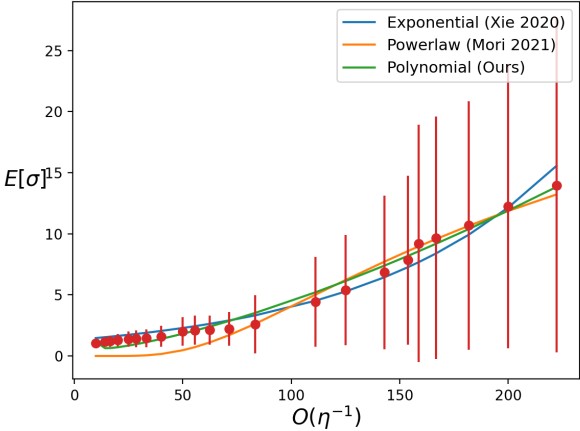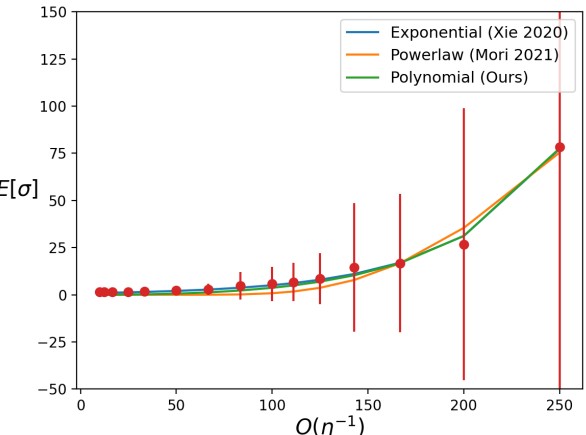

Figure 3: The mean escape time of SGD on Cardio (left) and Speech (right) datasets. The plots show the fitting base on three different methods: ours, Mori et al. (2021), and Xie et al. (2020). Each dot represent the mean escape time of 100 random seeds. We observe that in these experiments the theoretical fit provided by all works fits the empirical results relatively well.

calculating the noise of each parameter separately using multiple mini-batches; as opposed to Simsekli et al. (2019) that calculated the noise of multiple parameters on one mini-batch and averages over all parameters and batches to characterize the distribution of SGN. In Xie et al. (2020), the authors estimate SGN on a DNN with randomly initialized weights; we, on the other hand, estimate the properties of SGN based on a pre-trained DNN. Expressly, since we want to estimate the escape time, we reason that a pre-trained DNN would better characterize this property.

We examine the SGN of four ResNet variants and a Bert-based architecture 2. The ResNets were examined using two datasets: CIFAR100 Krizhevsky (2009) and CINIC10 Darlow et al. (2018). Bert's SGN was examined using CoLA Warstadt et al. (2018) dataset. The full technical details appear in appendix A.1.1. We show qualitative and quantitative evidence for SGN's heavy tail nature, the qualitative results are depicted in Fig. 1, and more plots are available in Sec. I.1.1. Furthermore Fig. 5 shows the $\alpha_i$ values of randomly sampled parameters. In this figure, if the noise of the sampled parameter were Gaussian, we would expect all the blobs to concentrate around $\alpha = 2$ (since at this value $S\alpha S$ boils down to a Gaussian distribution). The quantitative results depict the fitting error of the empirical distribution of SGN with Gaussian and $S\alpha S$ distributions. The fitting errors for ResNets on CINIC10 Darlow et al. (2018) are shown in Tab. 3, the full tables can be seen in Sec. I.1.2, for Cifar100 Krizhevsky (2009), the results are depicted in Tab. 1, and for Bert see Tab. 2.

**Learning rate decay** This paragraph aims to demonstrate that the LRdecay's effectiveness may be due to the attenuation of SGN. We show two experiments, first we trained ResNet110 He et al. (2015) on CIFAR100 Krizhevsky (2009), on epoch 280 the learning rate is decreased by a factor of 10. Fig. 2 shows that the learning rate decay results in a lower noise amplitude and less variance. In the second experiment, a ResNet20He et al. (2015) is trained in three different variations for 90 epochs; the first variation had LRdecay at epochs 30 and 60, the second had a batch-size increase at epochs 30 and 60, the third was trained with the same learning rate and batch size for the entire training process, the results show almost identical results on the first two cases, (i.e., LRdecay and batch increase) reaching a top-1 score of 66.7 and 66.4 on the validation set. In contrast, the third showed worse performances reaching a top-1 score of 53. Smith et al. (2017) performed similar experiments to show the similarity between decreasing the learning rate and increasing the batch size; however, their purpose was to suggest a method for improving training speed without degrading the results.

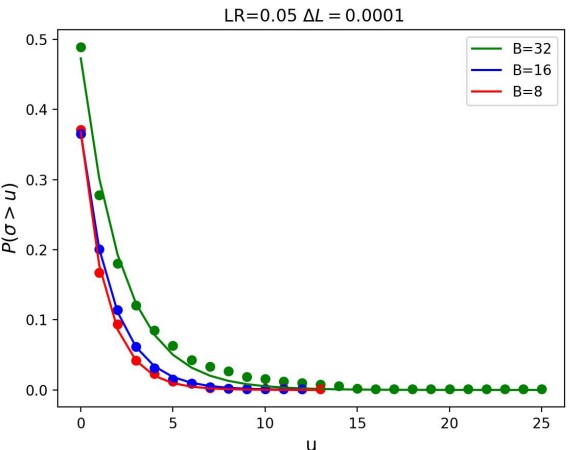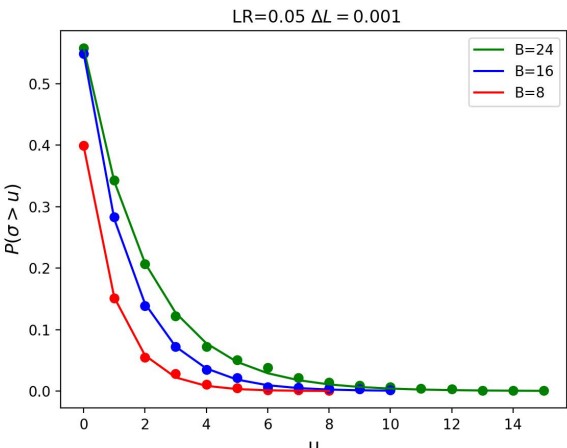

Figure 4: The x-axis represents the number of iterations. The y-axis represents the probability of exiting the basin; We the same model 1000 runs, in each run we kept the iteration of escaping the basin.The left plot shows results on Cardio dataset with different mini-batch sizes, the right plot shows the same on speech dataset.

LRdecay decreases the step size and the noise amplitude; on the other hand, increasing the batch size decreases the noise amplitude. Combining the results of the two experiments above, we may carefully deduce that the main effect of LRdecay is reducing the fluctuation in the gradient update phase and not decreasing the step size (step size is the movement of the deterministic process towards the minus of the gradient). SGN amplitude reduction enables the training process to get easier localization in the current promising domain.

**Different parameters hold different noise distributions?** This experiment shows that different DNN parameters lead to distinct SGN during training. We randomly sampled 100 parameters from five different DNNs. Then, we calculated the SGN and estimated $\alpha_i$ for each parameter; Fig. 5 depicts the results for the five DNNs. We observe that different parameters have noise that distributes differently during training. We can further notice that the variance is stretched on large segments of $\alpha_i$ values. This implies that building a framework that considers the DNN as one homogeneous system is insufficient; each parameter in the DNN has its characteristics, and we should consider this when modeling the noise. Models were trained using the same recipe as in A.1.1.

**Escape Time.** The following experiment validates Theorem 4.1. We trained a three-layer neural network with Relu activation on "Speech", "Ionosphere," and "Cardio" datasets Dua & Graff (2017). We first train the model using SGD with a learning rate of 0.05 and batch size of 128 until reaching a local minimum (see discussion Appendix B.1). After reaching the critical point, we decrease the mini-batch size and try to escape, Fig 3 shows the escape time using different learning rates. The escape time is measured by the number of iterations, averaged over 100 seeds. We fit empirical results to three theories, ours, Mori (1997) and Xie et al. (2020), all theories fitted with the same amount of free parameters. It can be seen that the empirical findings coincide with the three theories. Additional experiments appear in Appendix C.1.

**Probability of escaping after time u.** The following experiment validates Theorem E.4. We trained a three-layer neural network with Relu activation on Speech, Cardio and dataset Dua & Graff (2017) using SGD with a learning rate of 0.05 and batch size of 128 until convergence to a local minima. We measure the time to escape the local minimum on 1000 seeds and plot the probability distribution to exit as a function of time in Fig. 4.

**Escaping Axis** In this section, we demonstrate that the optimization process is more probable to escape from the axis with lower $\alpha_i$. We use a 2D Ackleys function; the escape process starts at the global minimum $\vec{0}$. We apply Gradient Descent with added $S\alpha S$ noise ($S\alpha S(\alpha_{x_1})$,$S\alpha S(\alpha_{x_2})$), where $\alpha_1 = \alpha_2 - \Delta$, learning rate of $1e - 4$, with no momentum or weight decay. Once the optimization process passes some predefined radius, we check which axis is larger. Fig 6 shows how probable it is to exit from $x_1$ based on 1000 different seeds. This result implies that as the $\delta$ between the $\alpha_i$s increases, the axis with the smaller value of $\alpha$ has more probability of being the axis through which the optimization process can escape.

## 6  Conclusions

Our experiments aid in validating that the $S\alpha S$ better characterized SGN visually and numerically. Furthermore, we showed that every parameter might evolve noise with distinct distribution parameter $\alpha$. We also presented experiments that support the claim that the main feature of LR schedulers comes from reducing the fluctuations of the SGN. Based on the mentioned experiments, we constructed a framework in $\mathbb{R}^N$ consisting of $N$ one-dimensional Lévy processes with $\alpha_i$-stable components. This framework enables us to better understand the nature of DNN training with SGD, such as the escaping properties from different local minima, a learning rate scheduler, and other parameters' effects in the DNN. Finally, we showed that parameters in the DNN that hold noise that distributes with low $\alpha_i$ have a unique role in the training process, helping the training process escape local minima.

**Limitations and Future Research** The presented framework is valid once the training process is near a local minimum; how the training acts in other states, for example, at the beginning of the training, is not intended to be solved in this work. The SDE approximation is true under small learning rate assumption. Further, how $\alpha$ evolves in time is still unclear and demands future research. It is also unclear why different parameters holds different SGN distributions. The SDE approximation is true under small learning rate, This work attempted to create a framework that model SGN as heavy tailed distribution in $R^N$ and further combine the effect of learning rate scheduler $s_t$, although we were able to reach mathematical estimations such as TheoremsE.4,4.3,4.3, we were not able to reach conclusions regarding the learning rate scheduler, and we hope future work will use our framework in order to further investigate this important aspect

### Broader Impact Statement

One of Deep learning main drawbacks is the lack of a fundamental theory, understanding this theory is critical for the advancement of the field. Surprisingly, the noise in SGD,is crucial in DNN optimization process, in this work we shed light on SGN distribution and effects on the training process, reveling a hand-breadth of it's mystery.

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
