# OpenReview forum: "Revisiting the Noise Model of Stochastic Gradient Descent"
_TMLR — Rejected by TMLR_

### Review · Reviewer_17rC · 2022-08-14

**Summary Of Contributions:**

The paper shows that different parameters in a neural network obey qualitatively different distributions.

While the authors also claimed quite a few other contributions, I remained unconvinced of those. See my review below.

**Broader Impact Concerns:**

Fine.

**Requested Changes:**

1. The authors need to answer to all my question in the weakness section
2. The variance of the related experiments needs to be reported
3. The figure font sizes are too small. The font size of the figures needs to match the font size of the main text
4. Figures have too low resolution. If the authors use pyplot, they should use at least >200 dpi for each figure
5. While I didn't point out explicitly, there are quite a few typos and unconventional English usage
6. The limitation of the work needs more discussion (e.g., when is the theory applicable?)


**Strengths And Weaknesses:**

In my opinion, the main strength of the paper is its attempt to understand the nature of the SGD gradient noise from a rather scientific point of view: build a minimal mathematical model and test its predictions. However, I feel unconvinced about many claims of the paper and I think the paper needs a significant revision before its contribution becomes clear.

Weaknesses. I feel that the authors made too many unsubstantiated claims. The experimental setting also does not feel too clear.
1. The result in table 1 and 2.
- The variance should also be reported to allow to actual scientific comparison
- Why are the resnets ordered in random order? For example, I think the authors should order the models as the depth increases (18, 34, 50, 110)
- Naively, I expect that as the depth of the ResNet increases, the errors (SSE/CHI2) should have a clear trend (or remain constant), but the present results show no trend at all. Why is this the case? To me, it feels that these variations are caused by the large variances of the reported value. This further emphasizes the need to report the variance. Still, even after the authors report the variance, the authors need to provide a clear explanation of why there is no clear trend.
2. Overclaim: "our experiments undoubtedly show that the $S\alpha S$ better characterized SGD visually and numerically."
- the word "undoubted" is too strong
- Why "visually"? I do not think any figure convinced me visually that the noise distribution is $S\alpha S$. For example, no reader looking at Figure 1 will be convinced: "Oh, this must an $S\alpha S$ distribution!"
3. Theorem 4.1 and its empirical demonstration in Figure 5 is weird.
- This result should be comparable to Theorem 3.2 in Xie. Why is the result of Xie not compared in this Figure?
- The authors should also discuss and compare with the result in proceedings.mlr.press/v162/mori22a.html
4. The experimental section claims to "use several different batch sizes and different learning rates." However, this is not true except for only table two
- Actually, I think the gradient distribution will change both qualitatively and quantitatively when using a very small or a very large learning rate and using a very small or large batch size. Do the results hold in these situations and why? The authors need to present and discuss these results
5. The experimental details are too vague
- The used batch sizes are not reported
- The used learning rates are not reported
- Even after such reporting, the authors need to explain why these particular values are chosen

---

> ### Author Response · Authors · 2022-09-26
> **Reply to reviewer 17rC**
>
> Part1:
>
>   (1) We agree with the reviewer; we added the standard deviation to the relevant tables.
>
>   (2) The ResNets models are sorted by order (based on the number of parameters), and ResNet110 is smaller than ResNet18. In [1], two types of architectures were presented: with and without bottleneck blocks.
>
>   (3) As the depth of the DNN increases, the ability to fit the noise should be worse, hence leading to a more significant error. We agree with this intuition, and ResNet34 does not follow this intuition. This might indicate that additional factors that influence the size of the error.
>
> Part2:
>
>    (1) We agree with the reviewer-"undoubtedly" is not the right word, and it does not reflect our scientific perspective. This sentence was rephrased.
>
>    (2) When analyzing the distribution of the SGN, it is essential to understand if the noise is characterized by Gaussian distribution or by some heavy-tailed distribution. We agree with the reviewer that it is more important to understand the heavy tail property of the empirical distribution than to mention a specific distribution.
>
> Part3:
>
>  We have rerun our experiments in section in order to compare with [2] and [3]; for all three methods, we fit two free parameters. The respective paragraph was also updated.
>
> Part4:
>
> Additional experiments with different batch sizes are now presented in Table 3 in the Appendix. For the results in this table, the estimation of the SGN is agnostic to the learning rate. This is because we are computing the gradient but not updating the weights; therefore, we did not conduct experiments with different learning rates.
>
> Part5:
>
>  We agree with this comment. We have added to the manuscript the complete details of the hyper-parameter choice.
>
> Extra point:
>
> We added a new discussion on the limitation of our work and future work. Please see p.12.

---

> > ### Comment · Reviewer_17rC · 2022-10-03
> > **thanks for the reply**
> >
> > After the rebuttal, I still feel reserved about the result of this work.
> >
> > The main argument is that a new distribution is more suitable for modeling SGD noise, but the advantage of this new model is satisfactorily demonstrated.
> >
> > For example, the results presented in current Figure 3 feel insufficient. The figure only shows a regime where all three theories agree very well. If the authors want to convince the readers that the proposed theory really advances the previous works, they should present a case where the three theories disagree in prediction, and the proposed theory fits the experimental result better. If the authors can present such a case for the experiment in Figure 3, I may be better convinced.

---

### Review · Reviewer_QKVm · 2022-08-28

**Summary Of Contributions:**

This paper extends the S$\alpha$S model for gradient noise proposed by Simsekli et al., 2019. In particular, this paper allow for unequal contributions from different parameters; see (3). Some theorems are proved and several numerical results are provided, but I do not get the insights they provide, regarding existing literature.

**Broader Impact Concerns:**

I don’t see any such concerns.

**Requested Changes:**

Indeed, I suggest direct rejection of this paper. That the draft seems not checked before submitted to TMLR shows lack of respect to TMLR.

If the paper has a chance of revision, then I suggest the following.
1. Please correct the spelling and grammar mistakes and improve the presentation.
2. Please clarify what more insights one can gain from this paper regarding existing literature.
3. Please check if there is any typo in (17).
4. Please fix Section B.2.
5. Please address my concern about the last claimed contribution as stated above.

**Strengths And Weaknesses:**

#### Strengths
1. The proposed model is novel. It is reasonable to consider a “real” multi-dimensional extension of the S$\alpha$S model of Simsekli et al., 2019; the latter considers the case where the noises in all dimensions are equally distributed.

#### Weaknesses
1. The presentation is awful. It seems that the draft was not checked before submitted. I would suggest rejection simply because of the presentation.
    1. There are a terrible number of spelling and grammar mistakes and even missing words. Please have the draft checked by any grammar checker.
    2. Please separate sentences by periods instead of commas.
    3. Please put the citations in parentheses when they are not used as nouns.
    4. What do the asterisks in Appendix B.2 mean?
    5. Some parts are imprecise. For example, does “local environment of a minimum” on p. 3 mean “a neighborhood of a minimizer”?
    6. Why is Figure 2 discussed later than Figure 4 and Figure 3 later than Figure 5 in Section 5?
2. I wonder what more insights one can gain from this paper, regarding existing literature.
    1. Surely the proposed model should fit better the empirical noise than the simpler S$\alpha$S model proposed by Simsekli et al., 2019, because the proposed model contains more parameters.
    2. The second observation at the end of p. 8 that the S$\alpha$S model allows faster escaping from local minima is already clear in Simsekli et al., 2019.
    3. I don’t see how (18) is obtained from (17), so the discussion below Theorem 4.3 becomes ineffective. I guess there is a typo in (17): The right hand side has no dependence on $i$.
3. The experimental setup on p. 10 reads unclear to me. I cannot even tell whether the “inaccuracy” in the numerical experiments in Simsekli et al., 2019 is avoided or not.
4. Section B.2: The validity of the local smoothness and strong convexity assumptions does not follow from existence of a convex neighborhood. Consider minimizing a constant function, for example.
5. The last claimed contribution, “showing that the effectiveness of the learning rate scheduler mainly evolves from noise attenuation and not step decaying,” does not seem to have a close relationship with the topic of the paper. Moreover, what is new regarding that Smith et al., 2018 had done the experiments?

---

> ### Author Response · Authors · 2022-09-26
> **Reply to reviewer QKVm**
>
> Part1:
>
> We agree with the reviewer that we had several typos and grammatical errors. The paper has been rephrased and revised to improve its presentation. We hope the reviewer can acknowledge that this was an honest mistake and by no means disrespecting the venue. The paper was recently sent to a professional proofreading service to improve the presentation further
>
> Part2:
>
> (1) This claim is not completely clear to us. First, Simsekli et al., 2019 [4] did not evaluate the quality of the SGN estimation; they only provided visual evidence to support the fit. Second, we agree with [2], that the methodology of [4] does not depict SGN; hence their empirical results are not comparable with ours. Third, we compared our results with [2] and [3] using the same amount of free parameters for all methods, p.11, "Escape Time" paragraph. Furthermore, our method mathematically expands [4] work to N dimensions and does not try to compete with [4].
> (2) Thanks for catching this mistake; we had a typo in eq.(18). This is now corrected in the manuscript.
>
> Part3:
>
> We rewrote the experimental setup on p.10 to clarify the details.
> For consistency with the results of Xie[2], we rely on code sent to us directly by the authors.
>
> Part4:
>
> This statement was indeed confusing. We rephrased the assumption in Section B2.
>
> Part5:
>
> Indeed, LRdecay is not the main result of our pape. Nonetheless, we wanted to present this example to demonstrate how impactful SGN is in the training process of a DNN. Our contribution in the LRdecay perspective is two folds: (i) Estimating SGN's norm before and after the LRdecay, and showing the attenuation of the noise and its concentration around zero. (ii) We built an $R^N$ framework that incorporates the learning rate scheduler and approximates the mean escape time, direction, and additional theorems that include $s_t$, i.e., the learning rate scheduler. Nevertheless, we must agree with the reviewer that our insights regarding the LRdecay mechanism are limited. This is now described in the limitation and future work section on p.12.
> Further Insights :
>
> (1) Our most important insight in the paper is to show that the SGN is in fact heavy-tailed. We do this using a more accurate experimental design compared with [2].
>
> (2) Different parameters in the DNN have different $\alpha$ values, i.e. mathematical model must incorporate this property, and it also raises the following question: why does one parameter have heavy-tail SGN while other parameters can be approximately Gaussian?
>
>
> [1]Kaiming He, Xiangyu Zhang, Shaoqing Ren, and Jian Sun. Deep Residual Learning for Image Recognition.
> arXiv e-prints, art. arXiv:1512.03385, December 2015
> [2]Zeke Xie, Issei Sato, and Masashi Sugiyama. A diffusion theory for deep learning dynamics: Stochastic
> gradient descent exponentially favors flat minima. arXiv e-prints, pp. arXiv–2002, 2020
> [3] Takashi Mori, Liu Ziyin, Kangqiao Liu, and Masahito Ueda. Power-law escape rate of SGD. arXiv e-prints,
> art. arXiv:2105.09557, May 2021
> [4]Umut Simsekli, Levent Sagun, and Mert Gurbuzbalaban. A tail-index analysis of stochastic gradient noise
> in deep neural networks. In International Conference on Machine Learning, pp. 5827–5837. PMLR, 2019

---

> > ### Comment · Reviewer_QKVm · 2022-10-10
> > **Re: Reply to reviewer QKVm**
> >
> > I appreciate the authors' effort to improve the presentation. Nevertheless, there are still many grammatical mistakes. In particular, there are many sentences separated by commas instead of periods for unclear reasons; there are many missing articles (a/an/the).
> >
> > There is no strong evidence supporting the new noise model. Table 1 and Table 2 suggest that the fitting errors with the SS model is smaller than those with the Gaussian model, but the error range is too large to support the claim; moreover, arguing the superiority of the S$\alpha$S model over the Gaussian model should be already done in existing work. In Figure 3, as Reviewer 17rC pointed out, all theories seem to work; the figure still does not provide a strong support for the authors' claim.
> >
> > The tables in Appendix I.1.2 may serve as evidences, if the tables compared the errors of the constant-parameter S$\alpha$S model and the proposed varying-parameter S$\alpha$S model (and make sure that the sample size is large enough to avoid overfitting---not in training the networks but evaluation of the fitting errors). The current tables compares the errors of the Gaussian and S$\alpha$S models, but existing work had argued for the superiority of the S$\alpha$S model.

---

### Review · Reviewer_vuAd · 2022-08-29

**Summary Of Contributions:**

This paper proposes a stochastic process S\alpha S Levy to depict the dynamics of SGD. The paper uses heavy stochastic differential equation to analyze. THe result helps better understanding of the training, e.g. learning rate decay.

**Requested Changes:**

I don't have arguments against the paper from professional perspectives. But I am curious about the implication of this S\alpha S Levy distribution, as the lemas and theorems are bit of non-intuitive. Could the authors discuss the effects of, e.g. the norm of the weights of each layer and what is their relations with those constants in the theorems.

**Strengths And Weaknesses:**

The paper studies a very important question in deep learning about how to characterize the minibatch sample variance the comes along with SGD. The product of this analysis gives an interesting results of the behavior nearby local minimum. Writings are professional and the introduction is well-listed. However I am not capable of checking the math derivation here.

---

> ### Author Response · Authors · 2022-09-26
> **Response to reviewer vuAd**
>
> Thanks for pointing this out; we will add intuitive explanations to make the lemmas and theorems more intuitive for the reader.
>
> Following this question, we conducted a new experiment (not included in the paper). In the new experiment, we evaluated the relation between the total norm of the weight and the mean escape time. The new experiment empirically demonstrated that models with higher weight norms were able to exit the basin much faster. These results raise interesting theoretical questions that may pave the way for future research. Since our results are preliminary, we decide to exclude them from the revised manuscript.

---

### Decision · Action_Editors · 2022-10-22

**Recommendation:** Reject

**Comment:**

As explained above, the advantage of using heavy tailed noise distribution in SGD can model the SGD behavior better. However, the advantage of the proposed parameter-wise SDE over existing methods is not well demonstrated, e.g., in Figure 3, the differences of the performance of the methods are almost the same. It is not clear in which case the proposed method can be better. To further improve the paper, evidence of the superiority of the proposed method should be proved, either theoretically or/and empirically.

Furthermore, the reviewers also found that the writing is not good enough, containing many grammar errors. Please carefully revise the paper for the next revision.

To sum up, due to the concerns raised by the reviewers, I propose rejection this time. But I also see potential significance of this paper if it can be revised accordingly. I encourage the authors to revise and re-submit for a second round review. Specifically, I'd like to see the following problems to be addressed in the next revision:
1. Differentiate the proposed work and related work (the current related work section is too vague to tell the difference between the proposed method and existing ones), especially I would like to see highlights of the differences.
2. Show the advantages of the proposed method with existing ones, either empirically or theoretically. For example, the third bullet in the summary contributions in page 2 claims the proposed method is better. It is not clear what it meant by "depict the training process better". To me, from the theoretical side, it derives some statistics of such processes such as the transition time and existing time of basins in Section 4. But these were not compared to the baseline of using a Levy process with the same parameter. So it is not clear whether the proposed method is better than the baseline. Empirically, some comparisons are provided with existing work, e.g., in Figure 3 to compare the mean escape time of SGD. But again, the results do not demonstrate "better" than existing methods. BTW, although I agree the noise distribution should not be Gaussian (light tailed), but what I read from Figure 2 is that the plotted distributions are light tailed instead of heavy tailed.
3. Carefully rewrite the paper and check for possible grammar errors. Although it is claimed the paper has gone through a professorial check, I still spot many errors. Please refer to the reviewer comments and revise the paper carefully.

**Audience:**

From the theoretical side, some audience might find some new results on the properties of the proposed noise in SGD, e.g., probability of existing a basin after some iterations. However, it is not quite clear whether this is better than existing methods. Empirically, there is no strong evidence that the proposed method is better than existing ones, as suggested by the reviewers.

**Claims And Evidence:**

The claim of modeling heavy tailed noise distribution in stochastic gradient descent is accepted, but there seems to be not enough evidence to support the advance of the proposed method compared to existing ones. Specifically, the superiority of SaS noise over the standard Gaussian noise has been studied by previous work. This papers proposes some extensions and develops related theory, but the empirical advantage of the proposed method over existing ones has not been demonstrated.